# Comparison of the Biomechanical Properties between Healthy and Whole Human and Porcine Stomachs

**DOI:** 10.3390/bioengineering11030233

**Published:** 2024-02-28

**Authors:** Feifei Li, Jiannan Liu, Xiaoyun Liu, Yaobin Wu, Lei Qian, Wenhua Huang, Yanbing Li

**Affiliations:** Guangdong Provincial Key Laboratory of Digital Medicine and Biomechanics, Guangdong Engineering Research Center for Translation of Medical 3D Printing Application, National Key Discipline of Human Anatomy, School of Basic Medical Sciences, Southern Medical University, Guangzhou 510515, China; lifeifei_dsa@163.com (F.L.); ljn980226@163.com (J.L.); 13822506902@163.com (X.L.); wuyaobin2018@smu.edu.cn (Y.W.)

**Keywords:** stomach tissue, uniaxial tension, biomechanical properties, hyperelastic, viscoelastic

## Abstract

Gastric cancer poses a societal and economic burden, prompting an exploration into the development of materials suitable for gastric reconstruction. However, there is a dearth of studies on the mechanical properties of porcine and human stomachs. Therefore, this study was conducted to elucidate their mechanical properties, focusing on interspecies correlations. Stress relaxation and tensile tests assessed the hyperelastic and viscoelastic characteristics of porcine and human stomachs. The thickness, stress–strain curve, elastic modulus, and stress relaxation were assessed. Porcine stomachs were significantly thicker than human stomachs. The stiffness contrast between porcine and human stomachs was evident. Porcine stomachs demonstrated varying elastic modulus values, with the highest in the longitudinal mucosa layer of the corpus and the lowest in the longitudinal intact layer of the fundus. In human stomachs, the elastic modulus of the longitudinal muscular layer of the antrum was the highest, whereas that of the circumferential muscularis layer of the corpus was the lowest. The degree of stress relaxation was higher in human stomachs than in porcine stomachs. This study comprehensively elucidated the differences between porcine and human stomachs attributable to variations across different regions and tissue layers, providing essential biomechanical support for subsequent studies in this field.

## 1. Introduction

Gastric cancer, with over one million reported new cases in 2020, emerges as the fifth most prevalent cancer worldwide, leading to 768,793 deaths and ranking fourth in terms of cancer-related mortality causes [1]. The incidence of gastric cancer is highest in Asia, particularly China [2,3]. Moreover, the estimated 5-year survival rate is significantly low, falling below 20% [2,4]. Owing to the rapid invasion and metastasis of gastric cancer cells, surgical resection emerges as the primary therapeutic intervention [5]. However, postoperative complications, such as gastric bleeding, anastomotic leakage, intestinal obstruction, gastroparesis, reflux esophagitis, postoperative infection, and dumping syndrome, significantly affect patients’ quality of life and pose safety threats [6]. Therefore, stomach tissue engineering has been proposed as a potential avenue to restore normal gastric mechanical and metabolic functions [7,8,9]. Identifying suitable materials for stomach repair or replacement could reduce the complexities and associated complications of current procedures. These materials may encompass synthetic materials or regenerative medicine; however, their efficacy relies upon the emulation of the mechanical properties inherent to the natural stomach. Therefore, an enhanced understanding of the mechanical properties of a normal, healthy, whole stomach is important in developing materials suitable for gastric reconstruction [10].

Experimental studies are essential to comprehensively understand the mechanical properties of the stomach. Two types of studies exist on the mechanical properties of the stomach: the first category encompasses whole-organ tests, wherein distensions are assessed at the organ level [11,12,13,14]; however, such experiments are limited owing to the lack of knowledge on the local properties of the tissues. The second category involves axial experiments conducted at the tissue scale. Additional local properties can be elucidated through tissue-level experiments conducted on stomach tissue strips using axial stretch [15,16,17,18,19,20] and compression [21,22] experiments. The nonlinear, viscoelastic, and anisotropic mechanical characteristics of the stomach tissue can be better elucidated using location-, tissue-level-, and orientation-related experiments. However, certain challenges persist in the execution of such experiments.

Currently, animal models are used in the majority of investigations on the mechanical properties of the stomach. Owing to the similarities in the shape, size, and function of porcine and human stomachs [16], as well as the ethical and practical constraints associated with obtaining human samples, porcine tissue is often used as a substitute for human tissue in medical research [23]. Therefore, experiments are often conducted on porcine stomachs [16,18,19,21,24,25]. However, despite the anatomical similarities, differences in body size and eating habits between pigs and humans raise questions regarding the suitability of the porcine model to accurately represent the mechanical properties of the human stomach. Therefore, further investigation is essential to elucidate the relationship between the porcine and human stomachs regarding their mechanical properties.

Compared to investigations on animals, there are very few mechanical tests on human stomachs. Moreover, most human stomach samples are obtained from obese patients following bariatric surgery [13,20,22]. Given that the body mass index (BMI) of these patients exceeds that of the general population, and that the resected samples only contain part of the fundus and corpus, excluding the entire antrum, it remains uncertain whether the findings of experiments involving obese patients align with those of individuals with a normal BMI. Moreover, further research is necessary to explore the mechanical properties of the antrum.

The aim in this study was to determine the mechanical properties of various anatomical regions, layers, and orientations in normal whole porcine and human stomachs, as well as to establish the relationship between porcine and human tissues. We also aimed to evaluate whether porcine tissue may serve as a substitute for the human stomach model by performing uniaxial tensile and stress relaxation experiments to compare their hyperelastic and viscoelastic properties. The findings of this investigation are poised to provide a basis for future research on artificial materials [26].

## 2. Materials and Methods

### 2.1. Sample Collection and Dissection

In this study, we utilized 18 porcine stomach specimens, which had an average weight of 93 ± 8 kg, from pigs approximately 6 months old. After being harvested, the stomachs were collected from a local slaughterhouse (Guangzhou, China) and brought to the laboratory in less than 30 min. The projected longitudinal (L_long_) and circumferential (L_circ_) lengths of the porcine stomachs were 252 ± 28 mm and 131 ± 16 mm, respectively, on average (Figure 1a). Subsequently, the stomachs were dissected along the greater curvature and washed with running water. The fundus, corpus, and antrum were identified based on the different colors of the gastric rugae and tissue structure (Figure 1b).

For the nine human stomachs included in this study, post-mortem collection was conducted on fresh frozen donor bodies stored at −80 °C [27] at the Department of Anatomy of the Southern Medical University, China. The average age of the nine donors was 50 ± 12 years; their details are outlined in Table 1. The average length and width of the human stomachs were 211 ± 14 mm and 133 ± 11 mm, respectively (Figure 1c). The tissue shape and color were used to define the fundus, corpus, and antrum (Figure 1d).

To keep the samples in a passive state, preventing their spontaneous contraction during testing, the organs were transported and stored in phosphate-buffered saline (PBS) (Biosharp, BL302A, Beijing, China) at 4 °C during the entire testing phase [28]. Hydrated samples were kept at 4 °C until testing, and all mechanical tests were completed within 6 h. The stomachs were emptied and measured in their deflated form before the samples were prepared. This study on human stomachs was approved by the Health Research Ethics Committee of Guangdong Provincial People’s Hospital on 21 April 2021 (approval number: KYZ202141702).

### 2.2. Mechanical Experiments

The stomach is a J-shaped reservoir of the digestive tract located between the esophagus and duodenum and has three main functions. These functions include acting as a storage vessel for food, facilitating the mixing of ingested food with digestive juices to form chyme, and regulating the release of digesta into the duodenum to ensure appropriate absorption and digestion [29,30]. Anatomically, the stomach can be divided into three major regions, the fundus, corpus, and antrum, arranged proximally to distally. Additionally, the stomach has two curvatures, the lesser and greater curvatures, which are concave to the upper right and lower left, respectively [31]. The mucosa, submucosa, muscularis, and serosa are the four layers that make up the multilayered composite structure of the stomach, ordered from the inside to the outside. The smooth muscle fibers in the muscular layer control the active behavior of the stomach, whereas the elastic fibers in the submucosa and muscular layer mainly control the passive behavior of the stomach [24,32].

Samples from the fundus, corpus, and antrum were used in the tests. Longitudinal and circumferential strips of the stomach wall were cut parallel and perpendicular to the greater curvature, respectively. The stomach wall was separated into left and right groups in order to ascertain the mechanical properties of the various layers. The left group was subjected to mucosal and muscular layer separation, whereas the right group remained intact (not separated) (Figure 2a,b). Ultimately, three regions, two orientations, and three layers were considered, resulting in 18 different cases. According to the tensile testing standard ASTM E8/E8M and the research protocols of previous studies [19], the strips used in this study were approximately 50 mm in length and 10 mm in width. Every sample was prepared and evaluated in a controlled setting with relative humidity of 60 ± 5% and a temperature of 20 ± 3 °C. In order to replicate their state in the body, PBS was sprayed on the stomach strip samples every 15 min during the tests.

A BOSE ElectroForce^®^ 3220-AT Series Ⅱ test instrument running the WinTest 7.2. software was used for the biomechanical test. The test device had a high-accuracy displacement sensor (±0.00001 mm) with a frequency response of 0–300 Hz and a 225 N load cell (±0.01 N) (Bose Corporation, ElectroForce Systems Group, New Castle, Delaware, USA). Each end of the tissue was gripped onto a fixture with a rough surface to hold the specimen. In an unloaded state, the sample was fixed between the two clamps. A Vernier caliper (DEGUQMNT-150T) was used to measure the thickness (*T_o_*) and width (*W_o_*) of the sample in the grip state, with accuracy of ±0.01 mm. The average of the three measured values were used to calculate the initial thickness and width. The initial test length (*L_o_*) of each sample was 20 mm, which was determined by the distance between the upper and lower fixtures. Nine markers were drawn at the centers of the samples and placed on a nine-dot grid to carry out the deformation-controlled experiments (Figure 2c). Images were acquired at a rate of 100 frames/s using a CCD camera (SONY FDR-AXP55) with a 3840 × 2160 pixel resolution. The displacement of the sample marker was photographed for subsequent dimensional analysis; the experimental process is illustrated in Figure 2d.

The mechanical testing process encompassed several steps. Prior to tensile testing, the biological tissues were subjected to preconditioning according to the viscoelastic properties. Ten loading–unloading cycles between 0 and 2 mm elongation (equivalent to 10% strain) were carried out at a rate of 0.2 mm/s as part of this preconditioning process. This procedure was performed thrice within the range of 0 to 5 mm elongation (equivalent to 25% strain). The material characteristics were analyzed during the last cycle [22]. After the last cycle, the samples were subjected to an instantaneous response of 25% strain at a deformation rate of 5 mm/s, which was followed by a 300-s rest period, in order to allow the nearly full development of stress relaxation phenomena.

### 2.3. Mechanical Data Analysis

The WinTest 7.2. software, which processes raw data from load cells, including time, load, and displacement, was used to calculate the stress and stretch ratios from the raw and morphometric data. Every dot in each image was subjected to real-time recording by a CCD camera, and the real-time coordinates were compared with the coordinates at the beginning of the experiment by a software program to calculate six real-time local longitudinal stretch ratios. Subsequently, the average stretch ratio for the test samples was calculated using the local stretch ratio [28]; see Figure 3a,b.

Owing to the large deformation of the stomachs during the experiment, the Cauchy stress description method was used to describe the mechanical properties of the stomachs more accurately. The volumes at the start of deformation (*o*) and the end of deformation (*d*) were considered to be equal for incompressible materials, which was applied to the Cauchy stress formula, expressed as follows:(1)$$\lambda_{ny}=\frac{d_{ny}}{o_{ny}}$$
(2)$$A_{o}=W_{o}T_{o}$$
(3)$$\;V=V_{o}=V_{d}=A_{o}L_{o}=A_{d}L_{d}$$
(4)$$A_{d}=\frac{A_{o}L_{o}}{L_{d}}$$
(5)$$\lambda =\frac{1}{6}\sum_{n=1}^{6}\lambda_{ny}$$
(6)σ=FAd=FAoLoLd=FλAo
where *λ* represents the stretching ratio, *n* denotes the number representing the specific position (Figure 3), *y* depicts the stretching direction, *d* indicates the deformed sample, *o* denotes the undeformed sample, A depicts the cross-sectional area, *W_o_* indicates the original width, *T_o_* depicts the initial thickness, *V* represents the volume, *L* indicates the length of the sample, *F* represents the load, and *σ* indicates the Cauchy stress.

The slope of the high-strain linear regions on the stress–strain curve was obtained by linear fitting and recorded as the elastic modulus (E); see Figure 3c.

### 2.4. Statistical Analysis

All data processing and statistical analyses were carried out using Microsoft Excel 2019, SPSS21.0, and GraphPad Prism 9. All data are expressed as the mean and standard deviation (X ± S). The Shapiro–Wilk test was used to assess the normality of all data. If the experimental data conformed to a normal distribution, they were subjected to two independent-sample t-tests and a one-way analysis of variance. However, when not conforming to a normal distribution, the Kruskal–Wallis test and Dunnett’s back-testing analysis were used for multiple comparisons. Furthermore, the Kruskal–Wallis test was applied, assuming unequal variance. *p*-values < 0.05 were considered statistically significant for all analyses.

## 3. Results

### 3.1. Morphological Data

The morphological data indicated significant differences in thickness between the porcine and human stomachs across the three locations and three layers (*p* < 0.001) (Figure 4). In the porcine stomachs, the intact stomach wall was significantly thinner in the fundus than in the corpus and antrum (4.14 ± 0.98 vs. 5.26 ± 0.75 vs. 5.17 ± 0.84 mm, *p* < 0.001). The thickness of the mucosa layer followed the sequence corpus > antrum > fundus (2.78 ± 0.53 vs. 2.00 ± 0.33 vs. 1.43 ± 0.34 mm, *p* < 0.001); however, that of the muscular layer followed the sequence antrum > fundus > corpus (3.70 ± 0.75 vs. 3.16 ± 1.00 vs. 2.59 ± 0.60 mm, *p* < 0.01). In the human stomachs, the thickness of the intact layer followed the sequence antrum > corpus > fundus (3.16 ± 0.71 vs. 2.36 ± 0.55 vs. 1.93 ± 0.61 mm, *p* < 0.05), whereas that of the mucosa layer followed the sequence corpus > antrum > fundus (1.23 ± 0.28 vs. 1.20 ± 0.35 vs. 0.98 ± 0.22 mm), with significant differences observed only between the fundus and corpus (*p* = 0.001). The thickness of the muscular layer followed the sequence antrum > corpus > fundus (2.06 ± 0.51 vs. 1.49 ± 0.29 vs. 1.14 ± 0.38 mm, *p* < 0.001).

### 3.2. Hyperelastic Mechanical Properties

The stress–stretch relationships and elastic moduli of the porcine and human stomach samples are shown in Figure 5 and Figure 6, respectively. In the longitudinal direction, a notable difference in elastic modulus was observed between the human and porcine stomachs. Specifically, for the intact layer of the fundus, the elastic modulus of the human stomachs was significantly higher than that of the porcine stomachs (1359.627 ± 562.283 vs. 194.684 ± 85.430 kPa, *p* < 0.001). In the mucosa layer of the fundus, although the stress–strain curve indicated that the elastic modulus of the porcine stomachs surpassed that of the human stomachs, the more pronounced rise in stress in the human stomachs led to a significantly larger elastic modulus compared to that of the porcine stomachs (2295.195 ± 1420.580 vs. 1000.970 ± 413.529 kPa, *p* = 0.001). However, the corpus exhibited a different pattern, where the elastic modulus of the porcine stomachs was significantly higher than that of the human stomachs (7041.760 ± 3002.070 vs. 1916.618 ± 585.237 kPa, *p* = 0.001). For the muscular layer, divergent results were observed in the fundus and antrum. In the fundus, the elastic modulus of the human stomachs was significantly larger than that of the porcine stomachs (954.846 ± 789.187 vs. 149.872 ± 89.513 kPa, *p* < 0.001). However, in the antrum, the elastic modulus of the porcine stomachs was significantly larger than that of the human stomachs (5513.323 ± 1435.131 vs. 2729.321 ± 721.034 kPa, *p* = 0.008). In the circumferential direction, the elastic modulus of the porcine stomachs was significantly higher than that of the human stomachs in terms of the mucosa layer of the corpus (5442.763 ± 1774.694 vs. 1535.957 ± 819.184 kPa, *p* = 0.012). In other cases, the elastic modulus of the human stomachs was significantly larger than that of the porcine stomachs (*p* < 0.05).

### 3.3. Viscoelastic Mechanical Properties

The stress relaxation curves of the porcine and human stomachs are shown in Figure 7. The stress was normalized. The steeper the stress relaxation curve, the faster the decrease in tissue stress, whereas lower tissue-normalized stress corresponded to a higher degree of stress relaxation. In the longitudinal direction, the stress relaxation degree of the muscular layer of the fundus in the porcine stomachs was higher than that in the human stomachs. In other cases, the stress relaxation degree was higher in the human stomachs than in the porcine stomachs. In the circumferential direction, the stress in the muscular layer of the porcine stomachs decreased more rapidly in the corpus; however, the final stress relaxation degree of the human stomachs was higher than that of the porcine stomachs. Conversely, porcine stomach samples exhibited a higher degree of relaxation in the antrum. In the remaining conditions, the stress relaxation degree was higher in human stomachs than in porcine stomachs. On the whole, the stress relaxation degree of human stomachs was higher.

All morphological data, elasticity moduli, and statistical comparison results are presented in the Appendix A.

## 4. Discussion

In this study, we investigated the hyperelastic and viscoelastic mechanical properties of porcine and human stomachs using uniaxial tensile tests and stress relaxation experiments to elucidate the mechanical behavior of stomach tissues under substantial deformations. The presented data encompass original findings from mechanical experiments conducted on whole normal porcine and human stomach tissues.

### 4.1. Hyperelastic Mechanical Properties

The investigation of tensile hyperelastic properties revealed differences between porcine and human stomachs on the whole, with variations evident across regions, layers, and directions. This complexity persisted not only between pigs and humans but also within each species.

On the whole, we observed that the stiffness of the corpus and antrum was higher than that of the fundus in porcine stomachs, consistent with the results of other studies [19,22,24]. However, in the circumferential direction of the mucosa layer, the elastic moduli of the fundus and antrum were similar, and there was no significant statistical difference between them. In the human samples, contrasting stiffness patterns were observed, with the antrum exhibiting greater stiffness than the fundus and corpus in the muscular layer; however, the same pattern was not observed in the full and mucosa layers. This finding contrasts with that of a previous study where antrum samples exhibited higher tensile stiffness than the fundus and corpus across all three layers [22]. Another study demonstrated that the average stress values of various parts of the stomach, from highest to lowest, were in the central body, proximal body, distal body, fundus, and antrum [20]. These discrepancies may be attributable to the difference in the source of samples between the studies. Fresh and whole normal human stomach samples were used in this experiment, whereas human stomach samples from obese patients following bariatric surgery were used in the previous studies. In the comparison between porcine and human stomachs, a significant difference in stiffness was observed in the longitudinal direction of the corpus, where the stiffness of the porcine stomachs was significantly higher than that of the human stomachs.

In the porcine stomach samples, the mucosa layer in the corpus in the longitudinal direction exhibited the highest hardness. This finding is consistent with the results of other investigators [19]. In human samples, the mucosa layer was stiffer than the intact and muscular layers, but only in the fundus. This finding differs from that of a previous study, where the mucosa layer was stiffer than the intact and muscle layers in the fundus, corpus, and antrum [22]. The limited number of studies on the mechanical properties of different layers in the human stomach, coupled with the differing sample criteria between both studies, may have contributed to these discrepancies. Our study underscores the differences in hyperelastic properties between porcine and human stomachs.

### 4.2. Viscoelastic Mechanical Properties

In the overall assessment, the stress relaxation degree of human stomachs surpassed that of porcine stomachs. The stress relaxation degree of the stomach wall exhibited apparent regional and layer heterogeneity, and this complexity differed between porcine and human stomachs.

For porcine stomachs, in the longitudinal direction of the intact layer, the stress relaxation degree of the antrum was higher than that of the fundus and corpus. However, no similar pattern was observed in the mucosa layer in any of these three regions. In the muscular layer, the stress relaxation degree of the fundus was higher than that of the corpus and antrum. In the circumferential direction, the stress relaxation behavior of the mucosa layer in the corpus was lower than that in the fundus and antrum; however, no similar pattern was observed for the full and muscular layers in these three regions. Comparatively, another study involving stress relaxation experiments conducted on porcine stomachs reported differing results. Their results demonstrated that, regarding the intact layer samples, the degree of stress relaxation of the corpus was lower than that of the fundus and antrum; however, a similar pattern was not observed for the mucosa and muscular layers across these three regions [22]. This disparity might be attributed to variations in age, as their experiment involved 3-month-old pigs, whereas our study involved 6-month-old pigs, suggesting potential impacts of different growth conditions. Our results demonstrated that, for human stomach samples, with the exception of the longitudinal direction of the mucosa layer, the antrum exhibited a lower stress relaxation degree than the fundus and corpus, consistent with the findings of other studies [13,22]. The experimental samples of the other studies were both from patients after bariatric surgery without the whole antrum, which may be the reason for the difference in the mucosal layer between the present study and other studies.

The largest stress relaxation degree was observed in the muscular layer of the fundus samples in porcine stomachs. However, no similar patterns were observed in the corpus or antrum. This finding contradicts the reported results of a study where the relaxation degree was the largest in the mucosa layer and smallest in the muscular layer [16]. These disparities could arise from the differences in the experimental loading protocols of both studies and the small sample size of their study. Our results showed that, for human stomach samples, the stress relaxation degree was higher in the mucosa layer than in the full and muscular layers of the fundus; however, the same pattern was not observed in the corpus and antrum. In contrast, the mucosa layer has been demonstrated to exhibit a lesser degree of relaxation in all regions [22]. This discrepancy may be attributed to our use of fresh, whole, normal human stomach samples compared to the previous study, where samples were obtained from obese patients following bariatric surgery. The exploration of viscoelasticity in our study provides a basis for future research on artificial materials.

### 4.3. Anisotropic Mechanical Behavior

Overall, our investigation into normal, whole porcine and human stomachs indicated that the mechanical properties of the stomach were directional. The stress–strain curves and elastic moduli of porcine stomachs showed that, for the intact, mucosa, and muscular layers of the fundus, corpus, and antrum, the stiffness of the longitudinal strip was higher than that of the circumferential strip, consistent with the results of a previous study [19]. Notably, in this study, the fundus exhibited a unique characteristic: no statistically significant difference was observed in the elastic modulus of the muscularis between the longitudinal and circumferential directions. This phenomenon could be attributed to the 25% maximum strain of the experiment, with the tissue remaining in the low-tensile-modulus region. Compared to that of porcine stomachs, the mechanical behavior of human stomachs differed; the stress–strain curve indicated that the human stomachs were anisotropic, whereas the statistical results of the elastic modulus indicated no significant differences between the longitudinal and circumferential directions. This finding could also be attributable to the human tissue remaining in a state of low tensile modulus within the 0–25% strain range applied in this experiment. Our study provides a foundational understanding for future research on the relationship between the microstructure and mechanical properties of the stomach.

### 4.4. Limitations

This study had certain limitations. First, obtaining human stomach tissues was difficult; therefore, our analysis involved stomach samples from only nine donors. Expanding the sample size could facilitate the assessment of the correlation between the mechanical properties of the stomach and factors such as sex, age, and BMI. Second, our focus was to investigate the passive uniaxial stretching behavior of human and porcine samples. Therefore, potential differences in the results of the uniaxial stretching performed in two different directions and those of biaxial testing may require further investigation.

## 5. Conclusions

In conclusion, we performed experiments on fresh, normal, and whole porcine and human stomachs, exploring different regions, layers, and directions. We characterized the hyperelastic and viscoelastic mechanical properties of the porcine and human stomachs using tensile and stress relaxation tests. The stiffness disparity between the porcine and human stomachs varied by region and layer, and the human stomach exhibited a greater degree of stress relaxation. The results in terms of the thickness, stress–strain curves, elastic modulus, and stress relaxation highlight the regional and layer-based heterogeneity of the stomach. The presence of anisotropy was also observed. These findings hold substantial significance, enhancing the understanding of the properties of the stomach and establishing a foundation for further research.

## Figures and Tables

**Figure 1 bioengineering-11-00233-f001:**
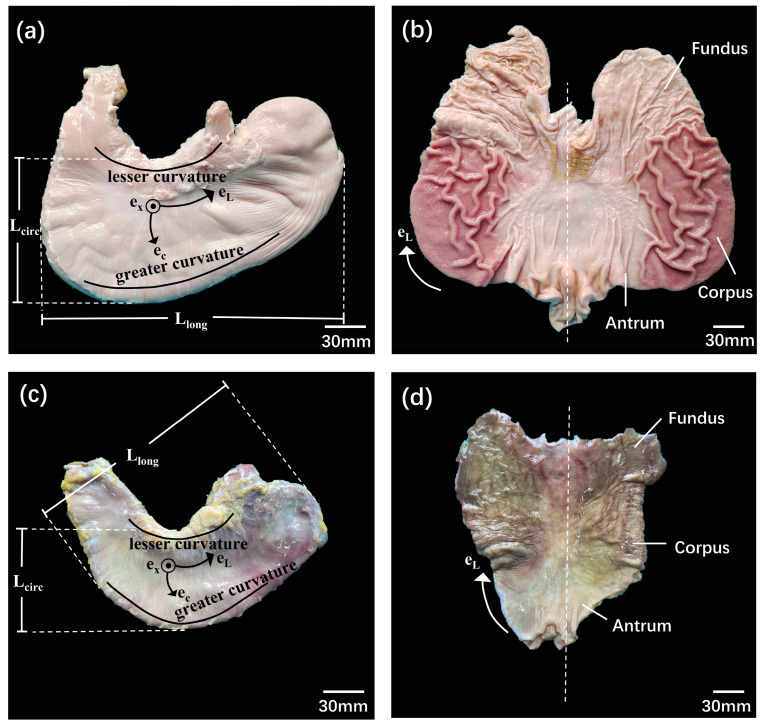
Anatomy and tissue sample dissection of porcine and human stomachs. (**a**) An external view of a deflated porcine stomach used to estimate its gross dimensions: longitudinal (L_long_) and circumferential (L_circ_) lengths. (**b**) An inside view of a porcine stomach that has been opened along its larger curvature, showing original colors for the optical identification of the different regions. (**c**) An external view of a deflated human stomach used to estimate its gross dimensions: L_long_ and L_circ_. (**d**) An inside view of a human stomach that has been opened along its larger curvature.

**Figure 2 bioengineering-11-00233-f002:**
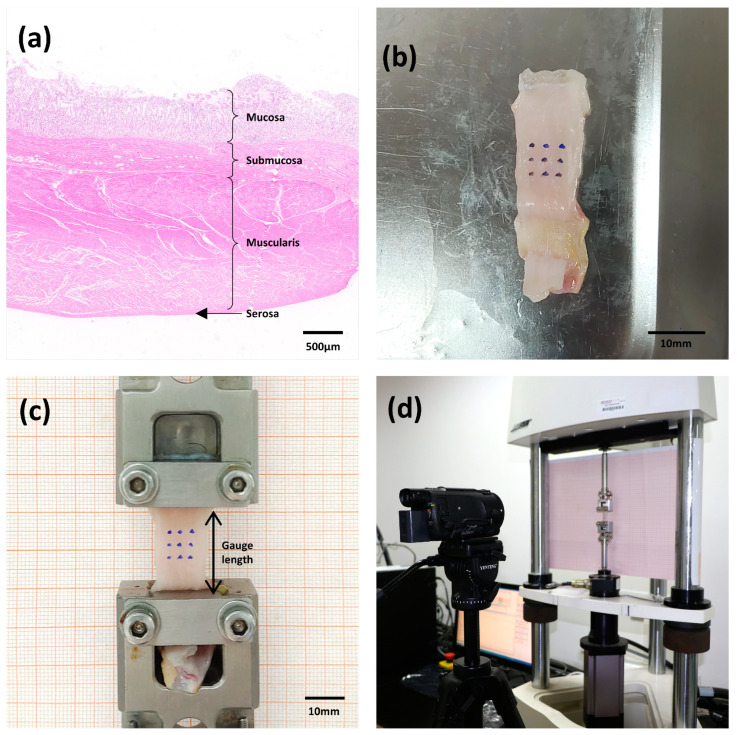
(**a**) Diagram of the layers of the porcine stomach wall (antrum) stained with hematoxylin and eosin. (**b**) Rectangular porcine stomach tissue specimens marked with nine tracking points. (**c**) View of the specimen mounted in the uniaxial testing machine. (**d**) General configuration of the setup for the uniaxial test.

**Figure 3 bioengineering-11-00233-f003:**
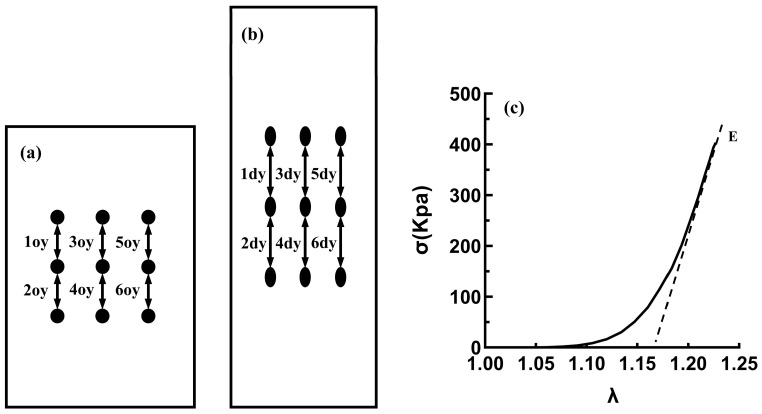
Schematic diagram of image strain data analysis. (**a**) At the beginning of the test. (**b**) At the end of the test. (**c**) The analysis diagram of the elastic modulus (E).

**Figure 4 bioengineering-11-00233-f004:**
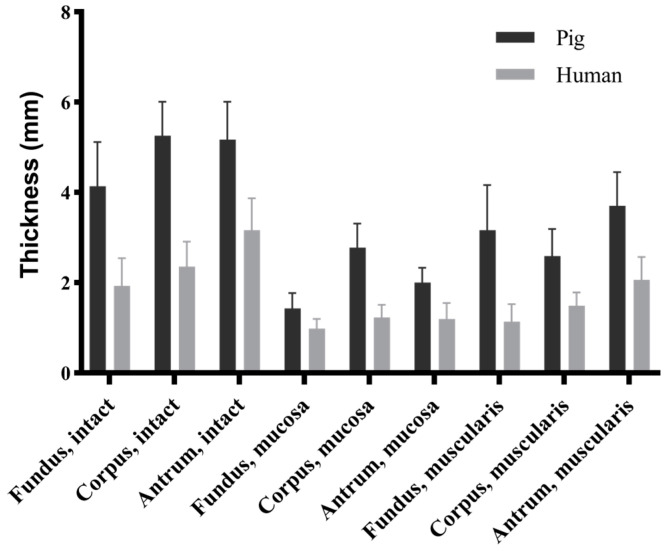
Thickness comparison of porcine and human stomach samples across each layer and region. All data are presented and mean ± standard deviation. Statistical differences were observed between pigs and humans across different regions and layers (*p* < 0.001).

**Figure 5 bioengineering-11-00233-f005:**
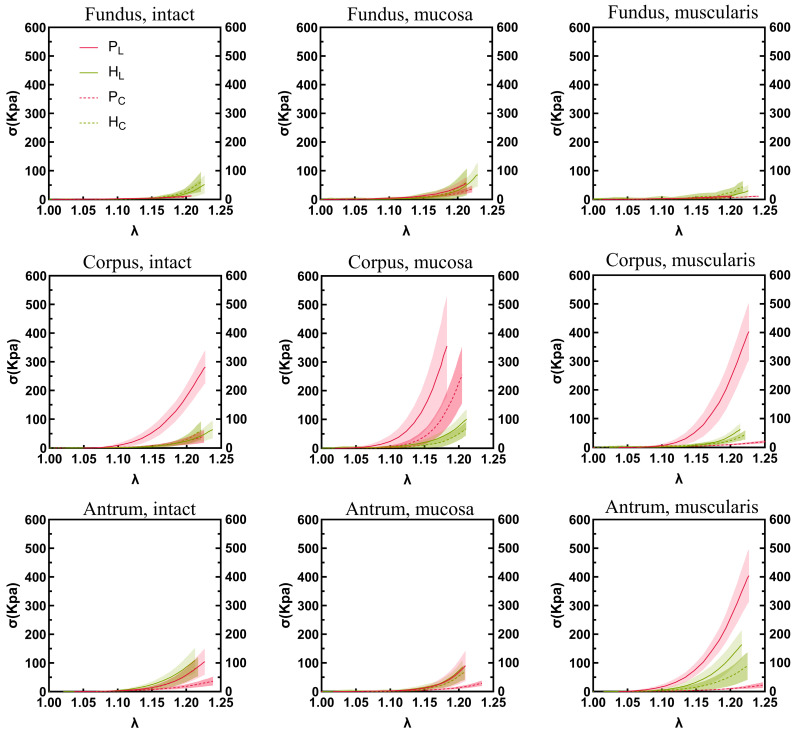
The stomach wall exhibits region-, layer-, and orientation-dependent stress–stretch behavior. Longitudinal and circumferential values are shown by solid curves and dotted lines, respectively. Standard deviations are shown as shaded regions. P, pig; H, human; L, longitudinal; C, circumferential.

**Figure 6 bioengineering-11-00233-f006:**
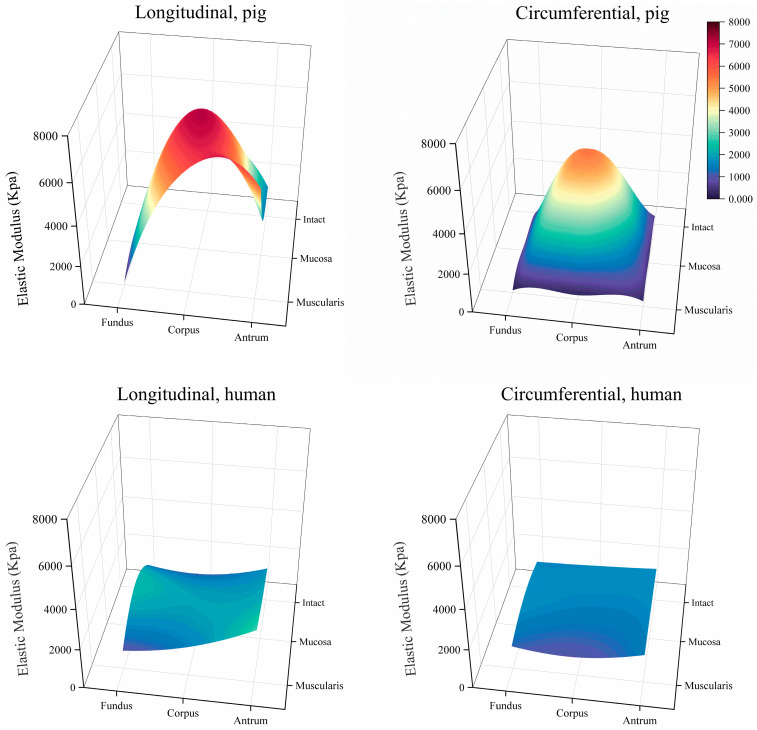
Elastic moduli for porcine and human stomachs.

**Figure 7 bioengineering-11-00233-f007:**
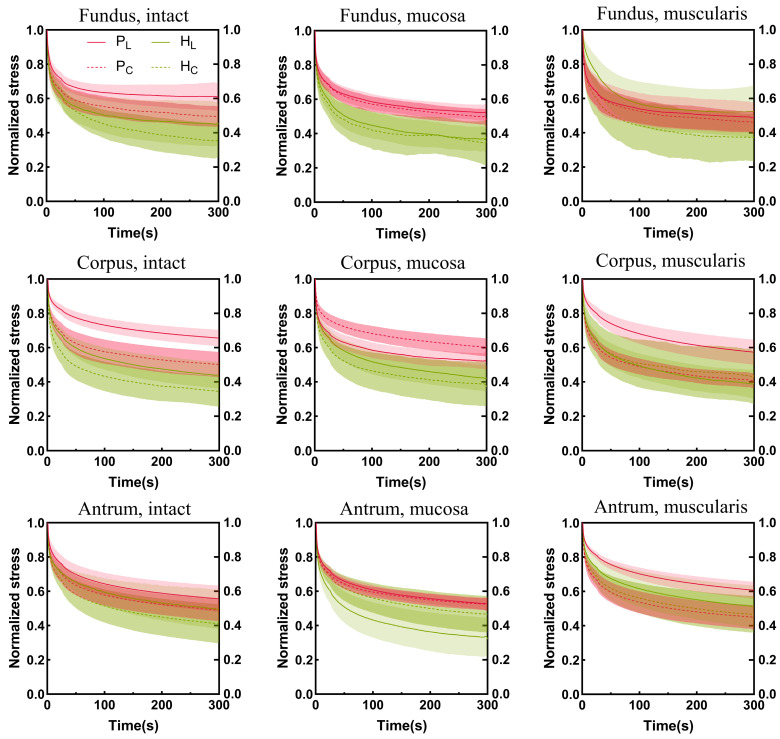
The stomach wall exhibits region-, layer-, and orientation-dependent stress relaxation behavior. The Cauchy stress is normalized. Longitudinal and circumferential values are shown by solid curves and dotted lines, respectively. Standard deviations are shown as shaded regions. P, pig; H, human; L, longitudinal; C, circumferential.

**Table 1 bioengineering-11-00233-t001:** Data of the human donors included in the study.

N	Sex	Age (Years)	Weight (Kg)	Height (m)	BMI (Kg/m^2^)
1	F	50	48	1.61	18.52
2	M	40	79	1.83	23.59
3	F	43	61	1.63	22.96
4	M	63	75	1.78	23.67
5	M	55	67	1.70	23.18
6	M	58	60	1.65	22.04
7	M	71	75	1.74	24.77
8	M	31	74	1.77	23.62
9	M	36	63	1.80	19.44
Mean ± SD		50 ± 12	67 ± 9	1.72 ± 0.07	22.42 ± 1.97

## Data Availability

The original contributions presented in the study are included in the article/Appendix A. Further inquiries can be directed to the corresponding authors.

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
