# Peer review of "Comparison of the Biomechanical Properties between Healthy and Whole Human and Porcine Stomachs"

_bioengineering, 2024, doi:10.3390/bioengineering11030233_

Round 1

Reviewer 1 Report

Comments and Suggestions for Authors

The authors aimed to address the lack of comprehensive understanding regarding the mechanical properties of both porcine and human stomachs, which is crucial for the development of materials used in gastric reconstruction procedures. They conducted stress relaxation and uniaxial tensile tests to evaluate the hyperelastic and viscoelastic characteristics of these stomachs, with a focus on comparing interspecies correlations. The paper is well written and structured, although there are some points that should be reconsidered.

Specific comments:

Abstract

Line 13. I’d suggest that “tensile tests” appear first in this sentence. 

1. Introduction

Lines 65-72. Authors should consider moving this paragraph to the discussion section or at least the idea if high BMI results align with normal BMI.

2. Materials and Methods

2.1 Sample collection and dissection

Lines 82-83. Are all the specimens the same age? Please rewrite this sentence.

Line 107. What do you mean with “passive state”?

2.2 Mechanical experiments

Lines 137-138. Does a temperature of 20 ºC replicate the body temperature?

Line 147. “The initial test length (L_o) of each sample was 20 mm” Did the authors applied an initial tension to the sample to find this initial length? It seems difficult for me to guarantee this exact value for all the samples.

2.3 Mechanical data analysis

Lines 172. Is this software able to compute the real time coordinates of the points in the images?

3 Results

3.1 Morphological data 

Line 228. The elastic modulus should be defined. At which region of the curve is obtained? Since the samples exhibited a nonlinear behavior this is an important point that authors should clarify.

Author Response

We would like to thank the reviewers for their careful reading, helpful comments, and constructive suggestions, which has significantly improved the presentation of our manuscript.

We have carefully considered all the comments and have revised our manuscript accordingly. The manuscript has also been double-checked, and the typos and grammar errors we found have been corrected. In the following section, we summarize our responses to each comment from the reviewers. We believe that our responses have addressed all the concerns from the reviewers. We hope our revised manuscript would be accepted for publication.

Case 1:

Abstract

Line 13. I’d suggest that “tensile tests” appear first in this sentence.

Response: Thank you for your suggestion. We have modified this sentence to "Stress relaxation and tensile tests assessed the hyperelastic and viscoelastic characteristics of porcine and human stomachs" and highlighted it on page 1, line 13.

Case 2:

  1. Introduction

Lines 65-72. Authors should consider moving this paragraph to the discussion section or at least the idea if high BMI results align with normal BMI.

Response: Thank you for your comment. This paragraph mainly describes the current research on the biomechanical properties of the human stomach. At present, most researchers use stomach tissues from people with obesity (high BMI), and few studies have been conducted using stomach tissues from people with normal BMI. Therefore, it is unclear whether there is a difference between stomachs from people with high or normal BMI, which is what we wanted to investigate. In the discussion section, the results of this study were compared with those of other studies, and it is indicated that there are some differences between the stomachs from the two kinds of people.

Case 3:

  1. Materials and Methods

2.1 Sample collection and dissection

Lines 82-83. Are all the specimens the same age? Please rewrite this sentence.

Response: Yes, the porcine stomachs used in this experiment were all from pigs approximately 6 months old. According to your suggestion, we have modified it to "In this study, we utilized 18 porcine stomach specimens, which had an average weight of 93 ± 8 kg, from pigs approximately 6 months old" and highlighted it on page 2, line 82.

Case 4:

  1. Materials and Methods

2.1 Sample collection and dissection

Line 107. What do you mean with “passive state”?

Response: To maintain a passive state means to prevent spontaneous contraction of the samples during testing. For better understanding, we have modified it to "To keep the samples in a passive state, preventing their spontaneous contraction during testing, the organs were transported and stored in phosphate-buffered saline (PBS) (Biosharp, BL302A) at 4 ℃ during the entire testing phase" and highlighted it on page 4, line 108.

Case 5:

  1. Materials and Methods

2.2 Mechanical experiments

Lines 137-138. Does a temperature of 20 ºC replicate the body temperature?

Response: Thank you for your comment. The experimental temperature used in this study was the ambient temperature at 20±3℃. The temperature was based on the results of previous studies, e.g., Sif Julie Friis used the ambient temperature of 23℃, Z.G. Jia adopted nominal room temperature of 20℃ in the study. and the experiment of R.C. Aydin was at room temperature 20℃.

Please see the references:

Friis, S.J.; Hansen, T.S.; Poulsen, M.; Gregersen, H.; Brüel, A.; Vinge Nygaard, J. Biomechanical properties of the stomach: A comprehensive comparative analysis of human and porcine gastric tissue. Journal of the Mechanical Behavior of Biomedical Materials 2023, 138, 105614, doi:https://doi.org/10.1016/j.jmbbm.2022.105614.

Jia, Z.G.; Li, W.; Zhou, Z.R. Mechanical characterization of stomach tissue under uniaxial tensile action. J Biomech 2015, 48, 651-658, doi:10.1016/j.jbiomech.2014.12.048.

Aydin, R.C.; Brandstaeter, S.; Braeu, F.A.; Steigenberger, M.; Marcus, R.P.; Nikolaou, K.; Notohamiprodjo, M.; Cyron, C.J. Experimental characterization of the biaxial mechanical properties of porcine gastric tissue. J Mech Behav Biomed Mater 2017, 74, 499-506, doi:10.1016/j.jmbbm.2017.07.028.

Case 6:

  1. Materials and Methods

2.2 Mechanical experiments

Line 147. “The initial test length (Lo) of each sample was 20 mm” Did the authors applied an initial tension to the sample to find this initial length? It seems difficult for me to guarantee this exact value for all the samples.

Response: Thank you for your questions. In this study, the samples were fixed on the upper fixture first, and dropped naturally under the action of gravity, and then fixed with the lower fixture, so no initial tension was applied to the samples. The initial test length (Lo) in this experiment is a fixed value, which is determined according to the distance between the upper and lower fixtures, as shown in Figure 2 (c), so that the initial test length of all samples can be consistent. We have modified it to "The initial test length (Lo) of each sample was 20 mm, which was determined by the distance between the upper and lower fixtures" and highlighted it on page 4, line 150.

Case 7:

  1. Materials and Methods

2.3 Mechanical data analysis

Lines 172. Is this software able to compute the real time coordinates of the points in the images?

Response: Yes, the software can calculate the dot displacement through the video image in real time. We modified the statement to "Every dot in each image was subjected to real-time recording by a CCD camera, and the real-time coordinates were compared with the coordinates at the beginning of the experiment by a software to calculate six real-time local longitudinal stretch ratios" and highlighted it on page 5, line 175.

Case 8:

3 Results

3.1 Morphological data

Line 228. The elastic modulus should be defined. At which region of the curve is obtained? Since the samples exhibited a nonlinear behavior this is an important point that authors should clarify.

Response: We have re-written this part according to the reviewer’s suggestion. We supplemented the calculation interval of the elastic modulus in article and annotated it in Figure 3(c). The supplementary content is "The slope of the high-strain linear regions on the stress-strain curve was obtained by linear fitting, and recorded as the elastic modulus (E), see Figure 3(c)" and highlighted it on page 6, line 200.

Thank you for your attention and time. We look forward to hearing from you.

Yours sincerely,

Feifei Li

23 Feb 2024

Southern Medical University

Reviewer 2 Report

Comments and Suggestions for Authors

This is an interesting manuscript and for me, it is worth publication. Some minor changes should be made before the acceptance.

1. Figure 1 - the labeling is too small, please enlarge.

2. Ethical approval - is it applicable for both humans and animals? This is not clear.

3. For the mechanical testing, are there any standards referred to? Eg: ISO, EC or others. Please include.

4. The sample size should be cited. Any references on the size?

5. The method for thickness measurement was not clearly explained.

6. Figure 5 is too small. For certain graphs, a scale from 400KPa and above can be removed. Please correct it.

7. I suggest including the results of maximum/failure load and ultimate strength. Those would be valuable for other researchers who are dealing with finite element analysis in the future.

Comments on the Quality of English Language

Minor grammatical errors.

Author Response

We would like to thank the reviewers for their careful reading, helpful comments, and constructive suggestions, which has significantly improved the presentation of our manuscript.

We have carefully considered all the comments and have revised our manuscript accordingly. The manuscript has also been double-checked, and the typos and grammar errors we found have been corrected. In the following section, we summarize our responses to each comment from the reviewers. We believe that our responses have addressed all the concerns from the reviewers. We hope our revised manuscript would be accepted for publication.

Case 1:

Figure 1 - the labeling is too small, please enlarge.

Response: According to your suggestion, we have modified Figure 1 and have enlarged the labels. Here are the revised results.

Case 2:

Ethical approval - is it applicable for both humans and animals? This is not clear.

Response: Thank you for the above suggestion. The human stomachs tissues used in this study were collected from the Department of Anatomy of the Southern Medical University, China. The study was conducted in accordance with the Declaration of Helsinki, and approved by the Health Research Ethics Committee of the Guangdong Provincial People’s Hospital on April 21, 2021 (approval number: KYZ202141702). Porcine stomachs were obtained from carcasses of pigs slaughtered for food purposes at a local abattoir, no animals were sacrificed solely for this research. Therefore, this study contains only one ethical approval. We have modified the statement to " This study on human stomachs was approved by the Health Research Ethics Committee of the Guangdong Provincial People’s Hospital on April 21, 2021 (approval number: KYZ202141702)" and highlighted it on page 4, line 113.

Case 3:

For the mechanical testing, are there any standards referred to? Eg: ISO, EC or others. Please include.

Response: For the mechanical testing, there is currently no standardized protocol for soft tissue biomechanical testing. We had reviewed several works on the mechanical tests of stomachs and found that the strain rate of 1% was generally adopted. Hence, we referred to the research method of Sif Julie Friis as the mechanical test scheme in our study.

Please see the reference:

Friis, S.J.; Hansen, T.S.; Poulsen, M.; Gregersen, H.; Brüel, A.; Vinge Nygaard, J. Biomechanical properties of the stomach: A comprehensive comparative analysis of human and porcine gastric tissue. Journal of the Mechanical Behavior of Biomedical Materials 2023, 138, 105614, doi:https://doi.org/10.1016/j.jmbbm.2022.105614.

Case 4:

The sample size should be cited. Any references on the size?

Response: Thank you so much for your suggestion. By reviewing the literature, we found that different researchers used different sizes of samples. We referred to the tensile testing standard ASTM E8/E8M, and the sample size of about 40 mm long and 8 mm wide was adopted by researcher Jingbo Zhao. Finally, our sample size was determined to be 10 mm in width and 50 mm in length. We modified the sentence to "According to the tensile testing standard ASTM E8/E8M and the research protocol of previous studies, the strips used in this study were approximately 50 mm in length and 10 mm in width" and highlighted it on page 4, line 136.

Please see the reference:

Zhao, J.; Liao, D.; Chen, P.; Kunwald, P.; Gregersen, H. Stomach stress and strain depend on location, direction and the layered structure. J Biomech 2008, 41, 3441-3447, doi:10.1016/j.jbiomech.2008.09.008.

Case 5:

The method for thickness measurement was not clearly explained.

Response: Thank you for your comment, we have revised the expression of thickness measurement indicating that the sample was in a clamping state during measurement, and the sentence has been modified to "A Vernier caliper (DEGUQMNT-150T) was used to measure the thickness (To) and width (Wo) of the sample in the grip state with an accuracy of ± 0.01 mm" and highlighted it on page 4, line 147.

Case 6:

Figure 5 is too small. For certain graphs, a scale from 400KPa and above can be removed. Please correct it.

Response: Thank you for your suggestion. In this study, 36 groups of stress-strain curves are summarized into Figure 5. Therefore, 600KPa is a large scale for some groups. However, we hope that Figure 5 can visually show the differences between the different groups, which is more convenient for readers to compare. According to your suggestion, we have enlarged Figure 5 from 12×12.75 cm to 17.73×18.84 cm.

Case 7:

I suggest including the results of maximum/failure load and ultimate strength. Those would be valuable for other researchers who are dealing with finite element analysis in the future.

Response: As the stress relaxation test was performed after the tensile test, the sample was not stretched to the maximum tensile length during the tensile test to ensure the stress relaxation test results. Therefore, the results of the maximum/failure load and ultimate strength could not be obtained. Thank you for your suggestion and we will conduct further research in future studies.

Thank you for your attention and time. We look forward to hearing from you.

Yours sincerely,

Feifei Li

23 Feb 2024

Southern Medical University

Reviewer 3 Report

Comments and Suggestions for Authors

This manuscript described the mechanical properties in terms of hyperelastic and viscoelastic comparison between porcine and human stomach tissues. This is a well-written and implemented manuscript. I do not have major issues with the contents of the manuscript. Some minor corrections are needed as follows:

1. Please recheck the use of significant numbers in reporting the result values in table 1.

2. Please recheck the use of the word "pig" in several places and figures which should be corrected to be "porcine".

3. Figures 5 and 7 are small and hard to see in detail. Revision is needed.

4. Apart from the curves in figures 5 and 7, I would think that the derived values showing the properties i.e stress relaxation etc. as tables for comparison would be useful and easier to evaluate.

Comments on the Quality of English Language

Recheck the writing error i.e "The In the overall assessment" and overall comprehension of the sentences.

Author Response

We would like to thank the reviewers for their careful reading, helpful comments, and constructive suggestions, which has significantly improved the presentation of our manuscript.

We have carefully considered all the comments and have revised our manuscript accordingly. The manuscript has also been double-checked, and the typos and grammar errors we found have been corrected. In the following section, we summarize our responses to each comment from the reviewers. We believe that our responses have addressed all the concerns from the reviewers. We hope our revised manuscript would be accepted for publication.

Case 1:

Please recheck the use of significant numbers in reporting the result values in table 1.

Response: Thank you for the above suggestion. Following your suggestion, we carefully checked the significant numbers in table 1. Considering the particularity of age and weight, and their units, they are expressed as integers. For the height, we modified it to keep it consistent with the BMI and kept two decimal places.

Table 1. Data of the human donors included in the study.

N0

Sex

Age (years)

Weight (Kg)

Height (m)

BMI (Kg/m2)

1

F

50

48

1.61

18.52

2

M

40

79

1.83

23.59

3

F

43

61

1.63

22.96

4

M

63

75

1.78

23.67

5

M

55

67

1.70

23.18

6

M

58

60

1.65

22.04

7

M

71

75

1.74

24.77

8

M

31

74

1.77

23.62

9

M

36

63

1.80

19.44

Mean±SD

50±12

67±9

1.72±0.07

22.42±1.97

Case 2:

Please recheck the use of the word "pig" in several places and figures which should be corrected to be "porcine".

Response: According to your advice, we have checked the article carefully and modified the expression from "pig" to "porcine" and highlighted it on page 2, line 82.

Case 3:

Figures 5 and 7 are small and hard to see in detail. Revision is needed.

Response: Thank you for your suggestion, we have enlarged Figures 5 and 7 from 12×12.75 cm to 17.73×18.84 cm in the article.

Case 4:

Apart from the curves in figures 5 and 7, I would think that the derived values showing the properties i.e stress relaxation etc. as tables for comparison would be useful and easier to evaluate.

Response: Thank you for your suggestion. At present, there is no standard for the presentation of stress relaxation in the field of soft tissue biomechanics. We referred to the data processing methods of other researchers, e.g, Ilaria Toniolo and Z.G. Jia, and normalized the stress relaxation results. Through processing, we could intuitively observe the speed and degree of stress decline in different groups, so as to compare the differences between different groups.

Please see the references:

Toniolo I, Fontanella CG, Foletto M, Carniel EL. Coupled experimental and computational approach to stomach biomechanics: Towards a validated characterization of gastric tissues mechanical properties. J Mech Behav Biomed Mater. 2022 Jan;125:104914. doi: 10.1016/j.jmbbm.2021.104914. Epub 2021 Oct 22. PMID: 34715641.

Jia, Z.G.; Li, W.; Zhou, Z.R. Mechanical characterization of stomach tissue under uniaxial tensile action. J Biomech 2015, 48, 651-658, doi:10.1016/j.jbiomech.2014.12.048.

Case 5:

Comments on the Quality of English Language

Recheck the writing error i.e "The In the overall assessment" and overall comprehension of the sentences.

Response: We apologize for the error and have modified the statement to “In the overall assessment” and highlighted it on page 11, line 321.

Thank you for your attention and time. We look forward to hearing from you.

Yours sincerely,

Feifei Li

23 Feb 2024

Southern Medical University

Round 2

Reviewer 1 Report

Comments and Suggestions for Authors

Authors have considered all my comments in the new version of the manuscript.